# Industrial scale high-throughput screening delivers multiple fast acting macrofilaricides

Rachel H. Clare [1], Catherine Bardelle[2], Paul Harper[2], W. David Hong[3], Ulf Börjesson[4], Kelly L. Johnston[1], Matthew Collier[2], Laura Myhill[1], Andrew Cassidy[1], Darren Plant[2], Helen Plant[2], Roger Clark[2], Darren A.N. Cook[1], Andrew Steven[1], John Archer[1], Paul McGillan [3], Sitthivut Charoensutthivarakul [1], Jaclyn Bibby[3], Raman Sharma[1], Gemma L. Nixon[3], Barton E. Slatko[5], Lindsey Cantin[5], Bo Wu[5], Joseph Turner [1], Louise Ford [1], Kirsty Rich[2], Mark Wigglesworth[2], Neil G. Berry [3], Paul M. O'Neill[3], Mark J. Taylor[1] & Stephen A. Ward[1]

Nematodes causing lymphatic filariasis and onchocerciasis rely on their bacterial endo-symbiont, *Wolbachia*, for survival and fecundity, making *Wolbachia* a promising therapeutic target. Here we perform a high-throughput screen of AstraZeneca's 1.3 million in-house compound library and identify 5 novel chemotypes with faster in vitro kill rates (<2 days) than existing anti-*Wolbachia* drugs that cure onchocerciasis and lymphatic filariasis. This industrial scale anthelmintic neglected tropical disease (NTD) screening campaign is the result of a partnership between the Anti-*Wolbachia* consortium (A•WOL) and AstraZeneca. The campaign was informed throughout by rational prioritisation and triage of compounds using cheminformatics to balance chemical diversity and drug like properties reducing the chance of attrition from the outset. Ongoing development of these multiple chemotypes, all with superior time-kill kinetics than registered antibiotics with anti-*Wolbachia* activity, has the potential to improve upon the current therapeutic options and deliver improved, safer and more selective macrofilaricidal drugs.

---

[1] Centre for Drugs and Diagnostics, Department of Parasitology, Liverpool School of Tropical Medicine, Pembroke Place, Liverpool L3 5QA, UK. [2] Hit Discovery, Discovery Sciences, IMED Biotech Unit, AstraZeneca, Macclesfield SK10 4TG, UK. [3] Department of Chemistry, University of Liverpool, Liverpool L69 7ZD, UK. [4] Hit Discovery, Discovery Sciences, IMED Biotech Unit, AstraZeneca, Gothenburg SE-431 83, Sweden. [5] Genome Biology Division, New England Biolabs, Inc, Ipswich 01938 MA, USA. Correspondence and requests for materials should be addressed to P.M.O'N. (email: P.M.Oneill01@liverpool.ac.uk) or to S.A.W. (email: Steve.Ward@lstmed.ac.uk)

Large-scale mass drug administration (MDA) programmes aimed at the control and elimination of the debilitating neglected tropical diseases (NTDs) onchocerciasis and lymphatic filariasis are hampered by the lack of a drug, which can safely kill the adult stages of the nematode parasites. The filarial nematodes which cause these diseases include *Wuchereria bancrofti*, *Brugia malayi* and *Brugia timori* for lymphatic filariasis whilst *Onchocerca volvulus* causes onchocerciasis[1,2]. These diseases afflict 157 million people worldwide and collectively are responsible for the loss of 3.3 million disability adjusted life years (DALYs) from the World's poorest communities[3,4]. Triple combinations of existing anthelminthic MDA drugs (ivermectin, diethylcarbamazine (DEC) and albendazole), have been shown to be superior to existing regimens for lymphatic filariasis in a phase II trial[5], but there are serious concerns in using these combinations for onchocerciasis and for lymphatic filariasis in areas of sub-Saharan Africa co-endemic for *Loa loa*. Concerns are due to severe adverse events (SAE) induced by rapid killing of *O. volvulus* microfilariae (Mf) in ocular tissues and Mazzotti reactions in the skin with DEC[6] and fatal encephalopathy in Loiasis following ivermectin, prompting the need for pre-screening procedures in at risk communities[7]. Fear of SAE in these communities are thought to contribute to poor-adherence to MDA in regions with prior experience of SAE[8,9].

An alternative treatment strategy, which avoids the risk of adverse events related to direct acting anti-filarial drugs, is to target the *Wolbachia* bacterial endosymbiont of the nematodes which cause onchocerciasis and lymphatic filariasis as it is vital for the nematodes' survival and fecundity[1]. Proof-of-concept field trials with the antibiotic doxycycline for 4–6 weeks, proves the value of this endosymbiont as a therapeutic target[10–16]. Depletion of this endosymbiotic bacteria was confirmed in these trials and resulted in the permanent sterilisation of the adult worms, blocking parasite transmission, followed by a slow innocuous death of the adult worm over 12–24 months. However, the duration of treatment with tetracyclines and contraindications in children and women of child-bearing age are barriers to the widespread scale-up of this approach. The A·WOL consortium was established to identify novel chemical starting points with anti-*Wolbachia* activity that could deliver shorter treatment regimens for these disabling filarial diseases[1,2,17].

The A·WOL consortium have developed and successfully utilised a fully validated whole cell screening assay using an insect cell line stably infected with *Wolbachia*, C6/36 (*w*AlbB)[18–20]. To facilitate the massive scale up of this screen from a capacity to screen thousands of compounds to an industrial standard high-throughput screen (HTS) capable of screening millions of compounds, A·WOL partnered with AstraZeneca's Global High-Throughput Screening (HTS) Centre. This collaboration established an industrial scale anthelmintic HTS for NTDs in their facility which had the capacity to screen the entire AstraZeneca 1.3 million compound collection.

## Results

**Overview of the primary HTS.** This screen incorporated a three-part assay summarised in Fig. 1. Initially C6/36 (*w*AlbB) cells (insect cells stably infected with the *Wolbachia* target) from a single cryopreserved batch were recovered over 7 days prior to plating into 384-well assay ready plates (containing the test compounds), using a semi-automated process allowing for daily batches of ~150 plates, set up 4 days per week for 8 weeks. After a 7-day incubation period with the test compounds the plates were moved into the second fully automated stage (using the Agilent Technologies BioCel system) involving formaldehyde fixation, DNA staining of the insect cell nuclei (Hoechst) for toxicity analysis and antibody staining specific to the intracellular *Wolbachia* (*w*BmPAL primary antibody and far-red secondary antibody). In the final stage the fixed and stained plates from stage 2 were processed through automated data acquisition (High Res

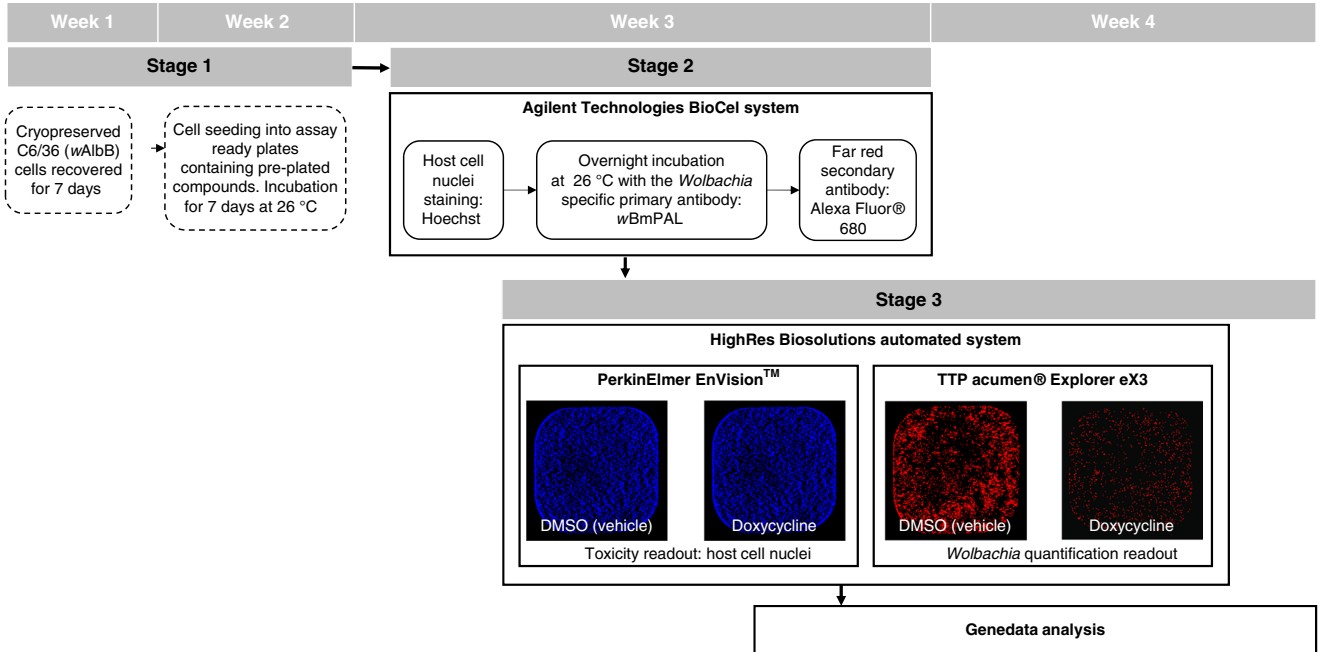

**Fig. 1** The high-throughput anti-*Wolbachia* screening flow. This extensive 16-step screen per batch across all three stages was completed over 3 weeks followed by overflow plate reading and data analysis in the 4th week when at full capacity of four daily batches of 150 plates per week. Manual processes are indicated by dashed boxes and fully automated steps in solid boxes. Bold rectangular boxes indicate the use of the fully automated systems: Agilent technologies BioCel system and HighRes Biosolutions automated system. The fluorescent images are representatives of the host nuclei (Hoechst) and *Wolbachia* (*w*BmPAL) staining taken from a whole well image (growth area of 10 mm$^2$) on the acumen® to demonstrate cells incubated with the vehicle/DMSO and doxycycline controls

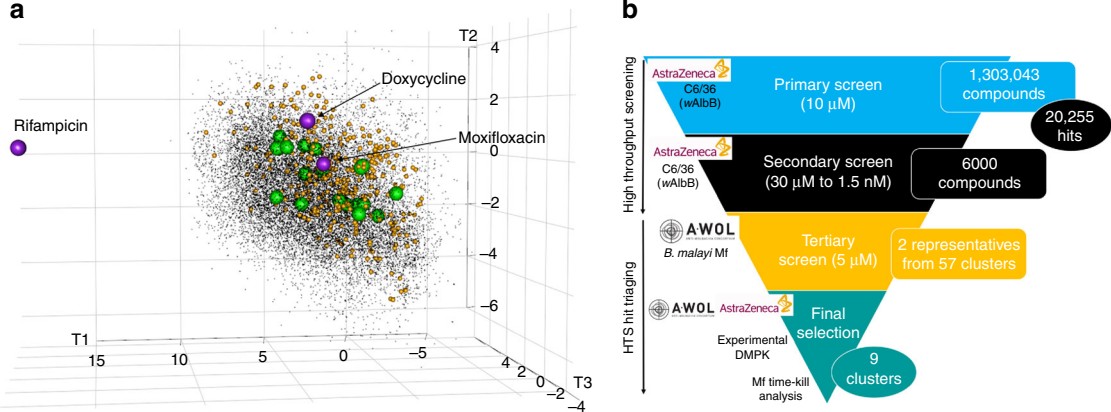

**Fig. 2** Overview of the HTS. **a** Results in ChemGPS chemical space. Purple spheres represent examples of current anti-*Wolbachia* agents; black points represent the ~20,000 actives identified in the primary HTS; orange spheres represent the 360 compounds from the 57 clusters tested in the microfilarial (Mf) assay; green spheres represent the 18 confirmed, potent hits. The position in ChemGPS space indicates a compound's size, hydrophobicity/hydrophilicity and flexibility (T1, T2 and T3 defined in ref. [21]). The hits span a large proportion of the chemical space covered by the ~6000 compounds. The 3D plot was produced in R using the rgl library (http://cran.r-project.org). **b** The full screening campaign and triaging process. Each stage includes the total compounds screened (rectangular box on the right) and hits identified (circular box on the right). The screen used (C6/36 (*w*AlbB) or *Brugia malayi* Mf) and screening location (AstraZeneca or A·WOL) are indicated on the left. Rational analysis was utilised at every stage based on cheminformatics and available data

Biosolutions system, incorporating EnVision™ and acumen® plate readers). This industry standardised assay was then used to screen AstraZeneca's full library of 1.3 million compounds within 10 weeks. In total this HTS included 30 × 3 week assays covering 3835 × 384-well plates. This primary HTS generated 20,255 hits defined as >80% reduction in *Wolbachia* with <60% toxicity to the host insect cell (eliminating false positives). This represents an overall hit rate of 1.56%. These compounds are depicted in chemical space in Fig. 2 (black points)[21].

Chemoinformatic analysis of the primary hits was used to identify the best ~6000 compounds for further evaluation in a secondary concentration response screen using the same HTS assay. Known antibacterials, pan assay interference compounds (PAINS), frequent hitters, known toxic compounds, compounds with an explosive risk, genotoxic compounds, reactive metabolites and hits with unwanted chemical groups were filtered out of the hit set[22–26]. The filtering rationale was based on prior knowledge and experience from both the AstraZeneca and A·WOL chemistry teams. The final selection was further triaged based on a balance of molecular weight, predicted logD, solubility, intrinsic clearance (models for human microsomes and rat hepatocytes) and chemotype diversity (details of compound selection, ranking and diversity assessment of the ~6000 compounds are provided in Supplementary Tables 1 and 2). A wide range of pIC50s was obtained from these ~6000 concentration response experiments, with 990 compounds showing a pIC50 > 6 (i.e. <1 μM IC50) against the *Wolbachia* target. In parallel these ~6000 compounds were also analysed for activity in a mammalian cell viability counter-screen to flag potential mammalian toxicity liabilities.

**HTS hit triaging: secondary and tertiary screening**. The ~6000 compounds tested in the secondary concentration response screen were clustered based on their chemical structures represented by ECFP6 fingerprints[27], with clusters of less than three compounds removed. The remaining clusters were prioritised based on manual assessment of anti-*Wolbachia* activity, mammalian toxicity, cluster size and chemical structure, as well as the measured compound purity. This process delivered 57 prioritised clusters containing 3–19 representatives and covering a total of 360 compounds. Figure 2 represents the chemical space

covered by these compounds (orange spheres). The tertiary prioritisation was carried out in a *B. malayi* Mf in vitro assay in order to assess the activity against the *Wolbachia* within a human filarial nematode, thus reducing attrition from issues, such as specificity to insect *Wolbachia*, indirect insect cell activity or barriers to drug penetration into nematodes. The two most potent representatives were selected from each of the 57 clusters for analysis. From the 113 available representative compounds tested at 5 μM, 85 showed less than 50% *Wolbachia* reduction (normalised to the doxycycline control). 11 compounds had between 50% and 80% *Wolbachia* reduction, while 17 compounds had >80% *Wolbachia* reduction. It is noteworthy that the confirmed hits spanned a large proportion of the chemical space covered by the original ~6000 compounds (Fig. 2). This diversity reduces the risk of attrition by aiming for differential targets.

**Discovery of five fast-acting chemical starting points**. Based on the anti-*Wolbachia* potency (cell and Mf) and in silico predictions of drug metabolism and pharmacokinetic (DMPK) properties, 18 compounds from 9 distinct clusters (two compounds in each cluster) were selected as the final hits for onward evaluation from this campaign (Supplementary Table 3). The criteria for compound selection were based on a selection score (derived according to Supplementary Table 4) with the resultant hits illustrated in chemical space in Fig. 3. We employed a simple ligand-efficiency metric, ligand efficiency-dependent lipophilicity index (LELP[28]), to aid in the prioritisation of the clusters balancing potency with lipophilicity. Notably, the calculated LELP for each of the highest scoring hits in each cluster is in the desirable range (≤10) further confirming the quality of the hit series identified.

To further validate the hit molecules from these 9 clusters, samples were sourced either from the AstraZeneca compound inventory or through re-synthesis. All 18 samples were characterised by nuclear magnetic resonance (NMR) and mass spectroscopy to confirm their chemical structures and by HPLC/LCMS to establish their purity (Supplementary Table 5 and Supplementary Data 1). These samples were also subject to confirmatory re-assessment in both the insect cell screen and Mf assays for anti-*Wolbachia* potency alongside confirmation of

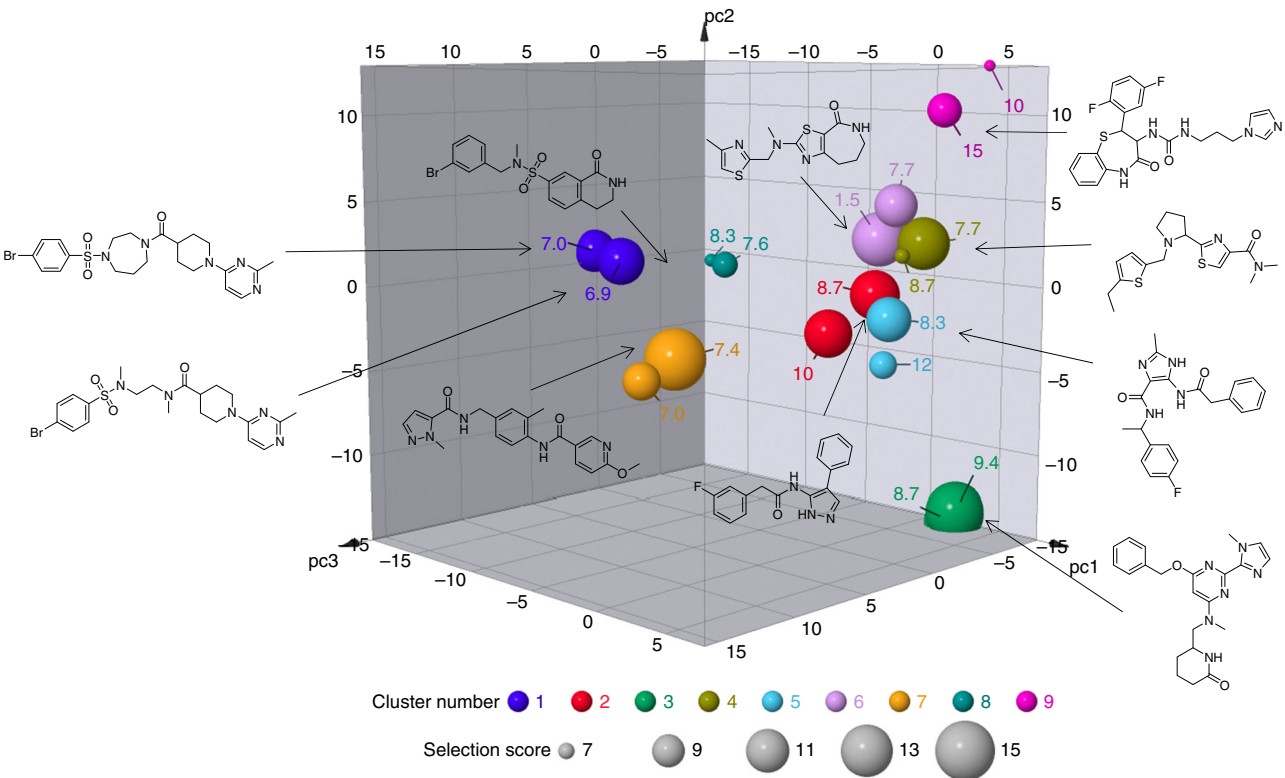

**Fig. 3** Eighteen selected hits represented in chemical space. The first three principal components (PC1–3) together account for 31.6% of the overall variance in the data set[39]. The chemotypes show clustering in chemical space. The size of each data point is proportional to the Selection Score. Selection Score is calculated by summation of the component scores using the criteria in Supplementary Table 3. The label for each point is the ligand efficiency-dependent lipophilicity index (LELP) with values <10 considered good[28]. A selection of the best compound structures is shown (large spheres) in a 3D plot produced by Datawarrior[39]. The cluster number is indicated by colour: cluster 1 (dark blue sphere), 2 (red sphere), 3 (green sphere), 4 (dark gold sphere), 5 (light blue sphere), 6 (lilac sphere), 7 (orange sphere), 8 (teal sphere), 9 (pink sphere)

DMPK properties (LogD$_{7.4}$, aqueous solubility in pH 7.4 PBS, human microsomal turnover, rat hepatocyte turnover and human plasma protein binding) (Fig. 4, Supplementary Figure 1 and Supplementary Table 4).

Based on these in vitro assessments and further scrutiny by A·WOL and AstraZeneca, the 9 hit series were placed into three categories, with each category containing representatives that have a similar balance between potency and DMPK properties. (Fig. 4, Category 3 hits are included in Supplementary Figure 1). From previous hit to lead optimisation studies strong anti-*Wolbachia* potency in the in vitro assays was found to be a reliable indicator of the potential of a chemotype to translate to acceptable in vivo anti-*Wolbachia* potency in the adult *B. malayi* mouse model[29,30]. Thus, this parameter was used as a key component for the following categorisation. Cluster 1, 2 and 3 were placed into Category 1 (Fig. 4), with all representatives expressing good potency in the Mf assay, superior to doxycycline in the same assay (EC$_{50}$ = 250 nM). In terms of DMPK properties, all Category 1 hit series have good physiochemical properties, i.e. aqueous solubility, but less than optimal metabolic stability. The balance of key properties for representative hits 1A–3A is depicted in radar plots (Fig. 4, desirable zone in shaded grey) and this representation identifies the potential metabolic (rat hepatocytes or human microsome) liabilities that need to be addressed moving forward.

Category 2 includes Clusters 4, 5 and 6 (Fig. 4). Representative pyrrolidinyl thiazole hit 4A (Cluster 4) has strong insect cell screen activity and good in vitro Mf potency. Solubility is also excellent for this molecule; however, the human microsomal

stability is poor and this feature should be targeted in optimisation studies following identification of the metabolic weak-spots through metabolite profiling studies. Representative imidazole carboxamide 5A (Cluster 5) has comparatively good Mf in vitro potency with lower solubility and moderate metabolic stability. Medicinal chemistry optimisation should focus on manipulation of one or both aromatic ring systems in this hit series to block P450 hydroxylation and should aim to reduce P450 interactions by lowering the overall lipophilicity. Radar plot analysis of 5A shows that this starting point has an excellent balance of all round properties that map onto the target feature set for hit prioritisation. Whilst cluster hit 6A has lower potency than representatives 4A or 5A, it does express very good in vitro human microsomal stability and acceptable solubility (Fig. 4). These features should enable a focused hit to lead optimisation on potency enhancement through expanding SAR around both the aromatic ring on the western side of the molecule, as well as functionalisation of the 5,6,7,8-tetrahydro-4H-thiazolo[5,4-c]azepin-4-one ring system.

The three remaining clusters 7, 8 and 9 have in vitro Mf potency in the range 1–5 μM (Supplementary Figure 1 and Supplementary Table 2). Pyrazole 7A has good measured solubility and high in vitro metabolic stability making potency-driven optimisation a primary focus in the hit to lead optimisation of this chemotype. Hit to lead optimisation of both 8A and 9A will likely be more challenging and will require multiparameter optimisation of Mf potency, solubility and microsomal stability.

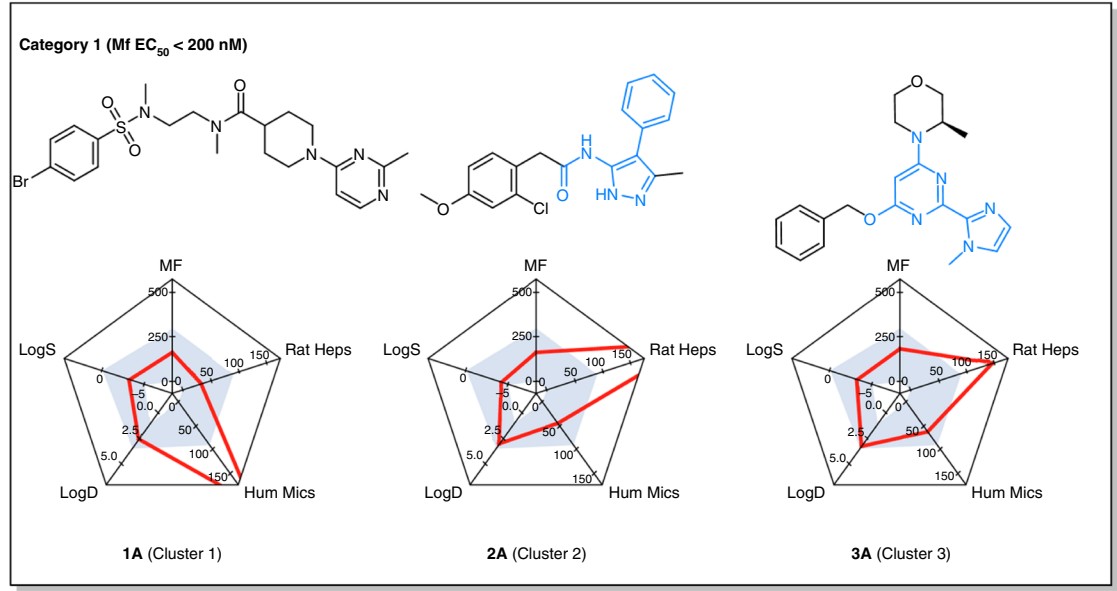

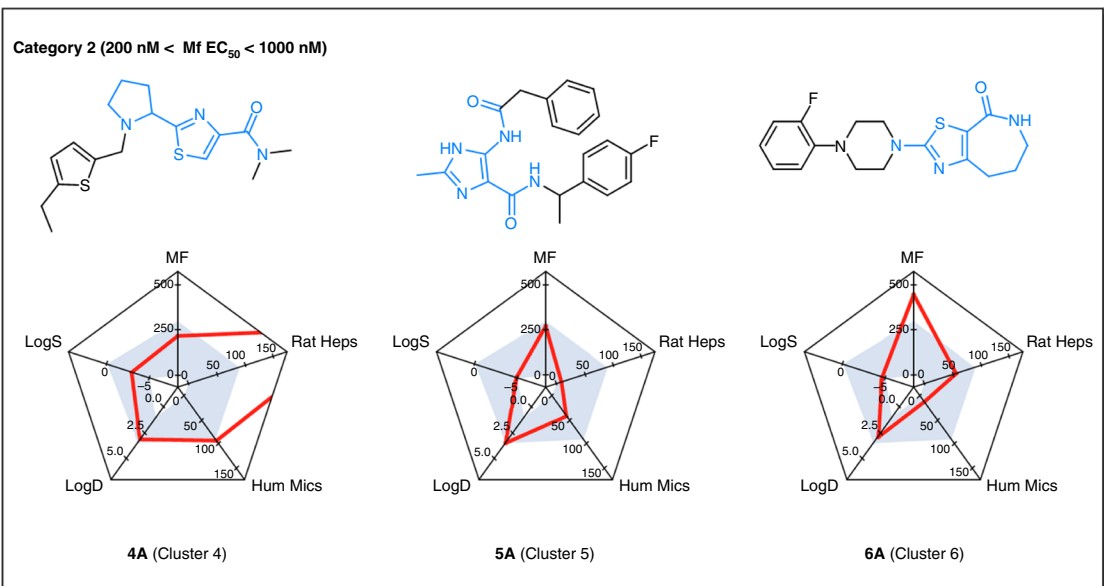

**Fig. 4** Anti-*Wolbachia* potency and drug-like properties of six cluster representatives. Nine hit series were categorised based on the balance between potency in the microfilarial assay and drug metabolism and pharmacokinetic (DMPK) properties. The figure shows the best representatives of the top six clusters. Five axes in the radar plots represent: $EC_{50}$ values (nM) in the anti-*Wolbachia* microfilarial assay (MF), rat hepatocytes clearance ($\mu l \, min^{-1}$ $1 \times 10^6$ cells$^{-1}$) (Rat Heps), human microsome clearance ($\mu l \, min^{-1} mg^{-1}$) (Hum Mics), LogD$_{7.4}$ (LogD) and log value of aqueous solubility in pH 7.4 PBS buffer ($\mu M$) (LogS). The core of each cluster chemotype is highlighted blue within the chemical structures

To compare the dynamics of the in vitro phenotype of our hits with existing anti-*Wolbachia* therapies, we employed a timed wash-out strategy (Fig. 5) at 10x $EC_{50}$ to establish the anti-*Wolbachia* time-kill kinetics of the selected chemotypes (1A–5A), using five replicates per compound. Time-kill kinetics of tetracyclines (doxycycline 1.21 μM and minocycline 0.9 μM) and rifampicin (0.2 μM) indicated that it takes >2 days to achieve a maximal *Wolbachia* reduction in this assay (70–90% when normalised to vehicle control). In contrast, all hit chemotypes (1A–5A) demonstrated this maximal activity (>70% *Wolbachia* reduction) after just 2 days exposure (ranging from 1.61 to 2.96 μM). Even more strikingly, after just a single day of exposure, doxycycline is inactive (9% *Wolbachia* reduction), yet chemotype 3A showed a 42% reduction and chemotypes 1A, 2A, 4A and 5A achieved *Wolbachia* reductions of 57–67%. This data presents a

dramatic temporal shift in the exposure time of <2 days to achieve a maximal *Wolbachia* reduction for these chemotypes compared to >4 days for known proof of concept anti-*Wolbachia* drugs.

The data indicates that these small drug like chemotypes mediate *Wolbachia* depletion much more rapidly than all existing anti-*Wolbachia* drugs. Optimisation and further development of any one of these five chemotypes has the potential to deliver molecules that could radically reduce treatment timeframes, which is one of the major limitations of the current policy drug, doxycycline (4–6 weeks regimens) for the treatment of onchocerciasis and lymphatic filariasis. Furthermore, it is accepted that time-kill kinetics is a fixed characteristic of a specific mechanism of drug action[31] and this data suggested these fast-acting hit chemotypes have different modes of action from those known anti-*Wolbachia* drugs.

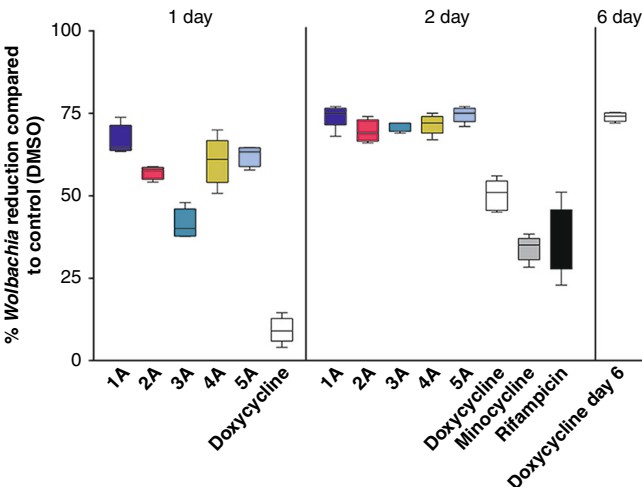

**Fig. 5** Anti-*Wolbachia* time-kill analysis. Representatives from the top five clusters (1A–5A), as well as established anti-*Wolbachia* antibacterials (doxycycline (white), minocycline (grey), rifampicin (black)) were assessed after exposure for 1 day and 2 days before washing and further incubation until the standard 6-day read out. *wsp:gst* ratio was normalised to the DMSO vehicle control resulting in a read out of mean percentage reduction from control. Doxycycline day 6 treatment represents the result for exposure to doxycycline at 5 µM for a full 6 days (no wash). The box extends from the 25th to the 75th percentile with the median presented by the central line. The whiskers are set at the minimum and maximum therefore indicating the range from five replicates. The cluster number is indicated by colour: cluster 1 (dark blue), 2 (magenta), 3 (teal), 4 (dark gold), and 5 (light blue)

## Discussion

The opportunity to access industry scale screening has delivered an enhanced portfolio of anti-*Wolbachia* chemotypes and identified 9 distinct chemical entities that represent excellent starting points for the discovery of novel drugs for the debilitating filarial diseases onchocerciasis and lymphatic filariasis. Our goals were achieved through a highly collaborative and multidisciplinary process synergistically building on the domain expertise complementarity of the academic and pharmaceutical groups. Working in partnership a workflow from assay validation, primary screen, to cheminformatic-driven chemotype prioritisation was achieved within 6 months (Fig. 2). The key component to the success of this work was the development of a robust model cell-based HTS, which through a 2 month screening timeframe maintained an average robust $Z'$ of 0.72 (0.06 standard deviation (SD) and 8.38% coefficient of variation (CV)) and average signal to background noise of 22.54 (2.77 SD and 12.27% CV) (Supplementary Figure 2). The primary screening of ~1.3 million compounds in 2 months represents a >25 fold increase in throughput compared to previous screens (25,000 to ~650,000 compounds per month)[18]. To put this in real terms, running the previous screens[18], even at full capacity, it would take over 4 years to complete the 1.3 million compounds described in this work. To date, our campaign is the highest throughput screen applied to anthelminthic NTD drug discovery ever undertaken, progressing not only the capacity from thousands to millions of compounds, but also increasing the number of hits and chemotypes by the same magnitude.

The 9 hit series identified in this campaign are not only new anti-*Wolbachia* chemotypes, but also novel antibacterial chemotypes. The only series to appear in the literature is Cluster 1; 4-piperidino pyrimidines, which are selective inhibitors of 2,3-oxidosqualene cyclase-lanosterol synthase in the cholesterol

biosynthesis pathway[32]. Intriguingly *Wolbachia* are known to perturb cholesterol homoeostasis in infected host cells[33].

In conclusion, we have described the first industrial scale anthelmintic NTD HTS campaign, a primary assessment of a 1.3 million compound library, a secondary concentration response screen of ~6000 compounds, and cheminformatic prioritisation and time-kill analysis to deliver a prioritised set of high-quality starting points for hit to lead optimisation efforts. The profiles of 9 prioritised hit chemotypes are described, with measured potency, physicochemical properties and DMPK analysis together with an assessment of the key areas of focus for onward medicinal chemistry optimisation studies. In vitro analysis of killing kinetics indicates that these fast-acting antibacterial agents have mechanisms of action different from any established anti-*Wolbachia* drugs which inhibit protein synthesis. Target identification studies of these fast-acting hits should reveal unexplored targets for not only the treatment of filarial disease but also generic antibacterial discovery.

## Methods

**Cell culture**. The mosquito (*Aedes albopictus*) derived cell line C6/36 (ATCC® CRL 1660), stably infected with *Wolbachia pipientis* wAlbB (C6/36(wAlbB)), was maintained in Leibovitz L15 medium containing 2 mM L-glutamine, 1% non-essential amino acids, 2% tryptose phosphate broth and 20% heat-inactivated foetal bovine serum (FBS) at 26 °C[18]. This screen exclusively used cells from the large-scale cryopreserved batch produced as part of this campaign by culturing the cells at scale before cryopreservation (90% FBS and 10% DMSO). This produced a cell bank of 190 × 1 ml cryovials containing 3 × 10^7 cells per vial.

The human monocytic THP-1 cell line (ATCC® TIB-202) used for mammalian cytotoxicity evaluation was routinely cultured in roller bottles. Growth media consisted of RPMI 1640 supplemented with 10% FBS and 2 mM L-glutamine. Cells were maintained at 37 °C with 95% humidity in 5% CO_2 in a shaking incubator. They were passaged every 2–3 days to ensure confluency did not exceed 1.5 × 10^6 cells/ml.

**Mammalian cell viability counter-screen**. All compounds selected for secondary IC_50 screen were tested in parallel for mammalian cell viability using the CellTiter-Blue® Cell Viability Assay. THP1 cells were removed from culture and plated at 1 × 10^4 cells/well (40 µl) into 384-well assay ready plates and incubated at 37 °C, 5% CO_2 for 48 h at 95% humidity. After the addition of CellTiter-Blue Viability reagent (8 µl of a 1:6 dilution) the assay plates were incubated for a further 2 h before fluorescence detected using a PerkinElmer EnVision™ plate reader.

**Wolbachia specific primary antibody production**. The antibody was made by Covance® by immunising five specific pathogen-free rabbits with *Wolbachia* peptidoglycan-associated lipoprotein from *B. malayi* (wBmPAL) produced and donated by New England BioLabs[34]. The antiserum was obtained and the anti-wBmPAL antibody purified by affinity column chromatography. From this we gained 350 ml of antibody.

**Primary HTS and secondary IC_50 screen assay set up**. On day 0, a single cryovial was recovered and the cells washed before being transferred to a T225 cm^2 flask. After a 7-day recovery *Wolbachia* infection level (quality control (QC)) was checked by SYTO®11 DNA staining of both the host cell nuclei and intracellular *Wolbachia*. The end-point read out from the Operetta® high content imaging system (Perkin Elmer) was the percentage of cells infected with *Wolbachia*. This was obtained from cytoplasm texture analysis by the associated Harmony® software[18]. Cells with a >50% of the population infected with *Wolbachia* were classed as having a good *Wolbachia* level and therefore used for screening through addition of 2000 cells per well into assay ready 384-well plates containing the test compounds. This resulted in a final screening concentration of 10 µM. Each plate included on-board controls; vehicle/DMSO (total/maximum *Wolbachia* signal) or doxycycline 5 µM (null/minimum *Wolbachia* signal), resulting in single shot screening of 352 compounds per plate for the primary screen. The secondary IC_50 screening plates were prepared in a similar way containing 36 compounds in a 10-point concentration range from 30 µM to 1.5 nM (1 in 3 serial dilution) in a serpentine pattern. QC plates were also created including full columns of the following controls; DMSO (total/max), 5 µM doxycycline (null/min), as well as 50 nM doxycycline (reference/medium *Wolbachia* signal). Daily batches contained ~150 compound plates in addition to the QC plates. Each compound plate was foil sealed before incubation for 7 days at 26 °C, 95% humidity and atmospheric air.

**Anti-*Wolbachia* and cytotoxicity HTS results characterisation**. After 7 days of drug incubation, the plates were loaded into the fully automated BioCel system which performed the following additions; fixation and DNA staining (1.8%

formaldehyde + 0.6 µg/ml Hoechst 33342, 20 min), permeabilisation (80 µl of PBS + 0.25% Triton-X 100, 30 min), blocking (PBS + 6% BSA, 40 min), overnight incubation in the *Wolbachia*-specific anti-*w*BmPAL primary antibody[34] (30 µl of 1 in 2000) followed by washing (PBS + 0.05% polysorbate) and a 1 h incubation in the secondary antibody (30 µl of 1 in 400), Alexa Fluor® 680 Goat Anti-Rabbit IgG (A-21076 Life Technologies™, far red fluorescence), before final washing (PBS + 0.05% polysorbate) and left in 40 µl of PBS with the plates foil sealed. Between each stage the wells were emptied of liquid.

The fixed antibody-stained plates were read on both a TTP acumen® (*Wolbachia* analysis) and a PerkinElmer EnVision™ (Hoechst host cell toxicity analysis) encompassed in two HighRes Biosolutions automated systems. The final reads out were; total *Wolbachia* area per well (gated for excessive area and fluorescence intensity, assumed to be artefacts) (TTP acumen®) and a single Hoechst fluorescent read out per well indicating the host cell density (PerkinElmer EnVision™). Per plate, the acumen® read was processed in ~20 min, whilst the EnVision™ read was obtained in 2 min. Using two acumen®s and the EnVision™, allowed for the 150 plate batch to be read in 28 h.

**HTS data analysis.** Genedata Screener software was used for all analysis, in which data for toxicity (Hoechst read on the EnVision™) and *Wolbachia* inhibition (antibody fluorescence read on the acumen®) were normalised on an individual plate basis. For the primary screen, the toxicity reads were normalised to the median read from all compounds on each plate as it is assumed the toxicity level should be limited and toxic compounds are randomly distributed over the plates in a given screening batch. The anti-*Wolbachia* reads were normalised to the maximum signal ('Max signal', DMSO controls) and the minimum signal ('Min signal', doxycycline 5 µM) these, respectively, representing 0% and 100% inhibition of *Wolbachia*.

The data was processed to identify 'hits' (*Wolbachia* inhibitors) that present little or no toxicity to the host cell line with the normalised *Wolbachia* data filtered at >80% inhibition. These active compounds were further annotated to identify those with >60% inhibition (i.e. toxicity) to the cell line. All plates were monitored for $Z$ prime, robust $Z$ score and signal to background.

**Primary HTS triaging.** Chemists from both A·WOL and AstraZeneca triaged the hit compounds from the primary screen, based on prior experience. Excluded compounds included known antibacterials, those previously worked on by A·WOL and to those with unwanted characteristics (e.g. PAINS, frequent hitters, known toxics, explosive risk and genotoxic or reactive metabolite substructures). The remaining compounds were prioritised based on a balance of molecular weight, predicted logD, solubility, intrinsic clearance (models for human microsomes and rat hepatocytes) and chemotype diversity. The selection was made by simultaneously (i) minimising the deviation from ideal ranges for each of the five physicochemical and DMPK properties considered and (ii) maximising the coverage of available chemotypes, using a Pareto multiobjective optimisation algorithm implemented in the PipelinePilot software.

**Secondary IC$_{50}$ screen triaging.** Fifty-seven clusters with at least three compounds each were selected for progression based on an overall manual assessment of anti-*Wolbachia* activity, mammalian toxicity, measured compound purity, cluster size, and chemical structure.

**Tertiary screening.** The two most potent compounds from the 57 prioritised clusters from the secondary screen triaging were screened in the microfilarial (Mf) assay at a final concentration of 5 µM. The Mf were obtained by peritoneal lavage, using a catheter, of gerbils (*Meriones unguiculatus*) harbouring a patent infection of *B. malayi*[35]. Mf were purified using a PD-10 desalting column (Fisher), centrifuged (311 RCF for 5 min at room temperature) before the addition of 8000 Mf/well of a 96-well plate (100 µl per well) in complete media consisting of RPMI supplemented with 10% FBS, 1% penicillin streptomycin and 1% amphotericin B. Compounds were diluted in complete media and 100 µl was added per well, providing a final concentration of 5 µM, with five replicates per compound. Each plate contained doxycycline (5 µM) and DMSO controls. After incubation at 37 °C for 6 days in 5% $CO_2$, visual assessment of Mf motility was performed and scored from 0 to 4 per well (where 0 = no movement and 4 = highly motile). DNA extraction per sample was performed using the QIAmp DNA Mini Kit before quantitative PCR was performed to obtain the ratio of *Wolbachia* Surface Protein gene copy number (*wsp*) to Glutathione S-Transferase gene copy number (*gst*)[36]. The DNA was amplified using the primers as indicated in Supplementary Table 6.

The *wsp:gst* ratio data allowed for normalisation of the *Wolbachia* level to the Mf biomass for each well. The change in *wsp:gst* for each compound compared to the DMSO control was calculated before normalisation to the onboard doxycycline control to allow for inter-assay comparison.

**Experimental DMPK screening.** The DMPK data used for triaging compounds, including LogD$_{7.4}$, aqueous solubility, plasma protein binding, microsome and hepatocyte clearance measurements, were generated using the standard AstraZeneca UK high-throughput platform[37,38].

## Data availability

The data that support the findings of this study are available on request from the corresponding author (S.A.W.) with permission from AstraZeneca. The compound data beyond that shared within the primary publication are not publicly available due to intellectual property restrictions.

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

## Acknowledgements

The A·WOL consortium is supported by grants from the Bill & Melinda Gates Foundation awarded to the Liverpool School of Tropical Medicine. This work was also supported by AstraZeneca. Antibody production was funded by NEB.

## Author contributions

R.H.C., P.H., C.B., M.C., R.C., D.P., H.P., K.L.J., D.A.N.C., L.F., N.G.B., U.B., W.D.H., P.M.O., J.T., M.J.T., S.A.W., M.W. and K.R. designed the research. R.H.C., C.B., M.C., L.M., A.C., P.H., A.S., J.A., N.G.B., U.B., P.M., S.C., B.S., B.W. and L.C. performed the research. R.H.C., C.B., M.C., N.G.B., U.B., W.D.H., P.M.O., G.L.N., R.S. and J.B. analysed the data. R.H.C., W.D.H., N.G.B., P.M.O., M.J.T., S.A.W. and C.B. wrote the paper.

## Additional information

**Competing interests:** C.B., P.H., U.B., D.P., H.P. and M.W. are employees and therefore shareholders of AstraZeneca, however, they have no financial or intellectual property rights to the structures presented in this manuscript which were transferred to the LSTM, a charitable organisation. The remaining authors declare no competing interests.

