## [Peer Review File · Nature Communications]

Reviewers' Comments:

Reviewer #1:

Remarks to the Author:

The manuscript by Clare et. al. takes advantage of a combination of cell-based high throughput screening approaches and chemo-informatics to identify novel anti-Wolbachia compounds. Targeting Wolbachia, an obligate symbiont of filarial nematodes, is a clinically effective means of killing adult nematodes that cause the neglected diseases onchocerciasis and lymphatic filariasis. The study describes an impressive high-throughput primary screen of 1.3 million compounds taking advantage of an insect cell line infected with Wolbachia and automated microscopy. Once these primary hits were identified, chemo-informatic approaches and mammalian toxicity studies narrowed the hits down to 6,000 compounds. Concentration analysis and cluster analysis resulted in 57 clusters containing 3-19 compounds per cluster. Representatives from each cluster were analyzed through in vitro microfilariae assays. This yielded a set of 17 compounds that with potent (>80% reduction) anti-Wolbachia activity. Additional informatic approaches and in vitro analysis identified 5 distinct chemotype clusters with anti-Wolbachia activity.

While the scale of the primary screen is impressive, a lack of follow-up in vivo studies diminish the impact of the manuscript. A number of high-throughput cell-based screens analyzing Wolbachia titer have been published thus reducing the novelty of this screen. A lesson from these previous small molecule screens is that only a fraction of the hits from the primary cell based screens translate into in vivo anti-Wolbachia activity in adult nematodes. It is likely this will be the case for the many of the compounds identified in this manuscript. This is in part due to the fact that there are significant differences between the genomes of insect and nematode Wolbachia and the two have very different life histories. At this point, the anti-Wolbachia drug discovery field has progressed to a point where adult nematode in vivo studies are essential for publication in a high impact journal.

Reviewer #2:

Remarks to the Author:

This manuscript describes a truly impressive effort to screen over one million compounds for new chemotypes that kill Wolbachia in an effort to combat various nematode parasites of humans. I have only two minor criticisms of the manuscript.

First, I find the manuscript too succinct with insufficient explanations for non-specialists, especially given the forgiving format of Nat Comm. For example, more explanation should be provided about the diseases being targeted, including the most prevalent nematode species, approximate numbers of people affected and DALYs numbers etc. In another example, much more background information should be provided on Wolbachia and its relationship with nematodes and why targeting Wolbachia is likely an effective approach to clear nematode infections. More background on most issues dealt with in the manuscript would help make the work more accessible to wider audience.

The second minor criticism I have is of the 'fast acting' claim. Its not clear from the figure or the text whether equivalent concentrations were used, or what the temporal dynamics of the compound's effects are. Its also not clear whether the compounds can completely rid the cells of Wolbachia, and over what time periods clearing can occur. Maybe the dynamics are such that a 70% reduction is seen quicker with the new chemotypes, but that 100% elimination happens quicker with dox for example. Also, how much clearing of Wolbachia is needed to compromise the nematode life cycle? Addressing these issues one way or another would help improve the manuscript.

Reviewer #3:

Remarks to the Author:

Onchocerciasis and lymphatic filariasis remain large scale and neglected diseases. The existing drugs have severe adverse events, cannot kill adult worms, and are difficult to use in eradication campaigns. The work disclosed herein is therefore important and timely. An alternate strategy to killing the worms is to kill an obligate endosymbiont of the worms: Wolbachia. This idea has clinical proof-of-concept with doxycycline, but that drug cannot be deployed for widespread use in children and women of child bearing potential due to adverse events. This work builds on that concept – it is therefore well validated and the group does an appropriate job of setting the context.

This work uses cell based screen with heterologous expression of target coupled with counterscreens to identify new leads targeting Wolbachia. Overall the work is executed at the state of the art and the conclusions presented within the manuscript are well supported by the presented data. The screening campaign Identified around 1000 compounds with potency better than 1 uM that were nontoxic to mammalian cells. The group then used standard methods to focus down on fast acting compounds with good drug like properties, ultimately Identified several that are faster acting than doxycycline, more potent, and selective. This is a key and important finding – suggesting that there are multiple opportunities for optimization of drug candidates from this foundation.

The work is well written and clearly presented; there are no significant changes that need to be made.

Reviewer # 1

The reviewer raises a question that ‘While the scale of the primary screen is impressive a lack of follow-up *in vivo* studies diminish the impact of the manuscript. A number of high-throughput cell-based screens analyzing *Wolbachia* titer have been published thus reducing the novelty of this screen. A lesson from these previous small molecule screens is that only a fraction of the hits from the primary cell based screens translate into *in vivo* anti-*Wolbachia* activity in adult nematodes. It is likely this will be the case for the many of the compounds identified in this manuscript. This is in part due to the fact that there are significant differences between the genomes of insect and nematode *Wolbachia* and the two have very different life histories. At this point, the anti-*Wolbachia* drug discovery field has progressed to a point where adult nematode *in vivo* studies are essential for publication in a high impact journal.’

In response we would argue that it is customary within drug discovery that these more complex and time-consuming *in vivo* screens are undertaken at the Lead Optimisation step rather than the hit identification step which is what this paper describes. Therefore, it would be unreasonable to expect *in vivo* data at this stage. As is common place in Lead Optimisation the compounds in this manuscript have been extensively triaged in order to remove compounds with known chemical liabilities to limit translation failures (*explained in the final paragraph on page 4*). Furthermore representative *in vitro* hits from the cell line were validated in the *in vitro* microfilarial assay where the *Wolbachia* genome is representative of that in human parasites, the concordance in these results is presented in *extended data table 4*. Of the 15 hit compounds from the *in vitro* cell line (EC50's <1µM) 60% (9/15) retained good activity in the Mf assay (EC50's < 0.5µM) with only 3 compounds failing to show activity (EC50's >5µM.) We have inserted the following lines on *page 5* to highlight this important point, ‘The tertiary prioritisation was carried out in a *Brugia malayi* microfilariae (Mf) *in vitro* assay in order to assess the activity against the *Wolbachia* within a human filarial nematode, thus reducing attrition from issues such as specificity to insect *Wolbachia*, indirect insect cell activity or barriers to drug penetration into nematodes.’ It is correct that in any discovery programme there is attrition as a molecule progresses from *in vitro* hit to an *in vivo* lead. This is usually due to issues related to drug access to the target (cellular penetration, drug disposition, transporter activity etc) and indeed attrition is expected. Our data to date indicates that all molecules where we have confirmed activity in both the insect and Mf *in vitro* assays always demonstrate some *in vivo* activity if we can achieve adequate exposures, as addressed to this effect on *page 7*, ‘From previous hit to lead optimisation studies strong potency in the *in vitro* assay was found to be a reliable indicator of the potential of a chemotype to translate to acceptable *in vivo* potency in the adult *Brugia malayi* mouse model^{29,30}.’

In response to the reviewer's comment on ‘A number of high-throughput cell-based screens analyzing *Wolbachia* titer have been published thus reducing the novelty of this screen’, we assume that the screens the reviewer refers to are the screening of a 2,664 compound library and 10,000 compound library published by Johnston et al. in 2014 and 2017 respectively. We would argue that the screen described in this manuscript is a paradigm shifting advance over these non-automated ≤10,000 compound libraries earlier screened. This manuscript describes the first industrial scale fully automated high throughput screening platform for filarial NTDs. The screen was capable of testing ~650,000 compounds/month in single point and greater than 6,000 compounds/month in full dose response compared to 25,000 compounds at single point per month using the predominantly manual based high content screen described by Johnston *et al.* (Johnston *et al.*, 2017) and 1,000 compounds per month screen described in (Johnston *et al.*, 2014). The industrial scale screen described in this manuscript has allowed the complete diversity set of AstraZeneca's

compound library to be evaluated for anti-*Wolbachia* activity (see 'Background' page 3 and 'Discussion' page 9/10 included below). To put this in context and as described on page 10, the approach described by Johnston *et al.* 2017 would have taken over 4 years to screen the library described in this manuscript.

'Background' page 3:

'To facilitate a paradigm shift in the scale up of this screen from a capacity to screen thousands of compounds to an industrial standard high-throughput screen (HTS) capable of screening millions of compounds, A-WOL partnered with AstraZeneca's Global High-Throughput Screening (HTS) Centre. This collaboration established the first industrial scale anthelmintic HTS for neglected tropical diseases in their facility which had the capacity to screen the entire AstraZeneca 1.3 million compound collection.'

'Discussion' page 9/10:

The primary screening of ~1.3 million compounds in 2 months represents a > 25 fold increase in throughput compared to previous screens (25,000 to ~650,000 compounds per month)¹⁸. To put this in real terms, running the previous screens, even at full capacity, it would take over 4 years to complete the 1.3 million compounds described in this work. To date, our campaign is the highest throughput screen applied to anthelmintic NTD drug discovery ever undertaken, progressing not only the capacity from thousands to millions of compounds, but also increasing the number of hits and chemotypes by the same magnitude.

In addition, the collaborative nature of this work in these NTDs is novel. It is the first time that drug discovery for lymphatic filariasis and onchocerciasis has been exposed to a pharmaceutical scale screening collection. This was enabled through a strong partnership between a pharmaceutical company and an academic consortium, which facilitated the development of this novel assay, screening as well as annotation and triage of the best compounds. Simply put, without this partnership this scale and breadth of screening for these diseases would not have been possible.

Finally, this work has resulted in the identification of 9 novel compound series of high quality through their highly promising depth of SAR, DMPK and fast acting anti-*Wolbachia* profiles. The latter of which has been identified for the first time through this work.

Reviewer #2

The reviewer has requested a more detailed overview , 'First, I find the manuscript too succinct with insufficient explanations for non-specialists, especially given the forgiving format of Nat Comm. For example, more explanation should be provided about the diseases being targeted, including the most prevalent nematode species, approximate numbers of people affected and DALYs numbers etc. In another example, much more background information should be provided on *Wolbachia* and its relationship with nematodes and why targeting *Wolbachia* is likely an effective approach to clear nematode infections. More background on most issues dealt with in the manuscript would help make the work more accessible to wider audience.'

In response we have added more detail as follows in sections 'Background' pages 2 and 3:

'The filarial nematodes which cause these diseases include *Wuchereria bancrofti*, *Brugia malayi* and *Brugia timori* for lymphatic filariasis whilst *Onchocerca volvulus* causes onchocerciasis^{1,2}. These diseases afflict 157 million people worldwide and collectively are responsible for the loss of 3.3 million disability adjusted life years (DALYs) from the World's poorest communities^{3,4}.

An alternative treatment strategy, which avoids the risk of adverse events related to direct acting anti-filarial drugs, is to target the *Wolbachia* bacterial endosymbiont of the nematodes which cause onchocerciasis and lymphatic filariasis as it is vital for the nematodes survival and fecundity¹. Proof-of-concept field trials with the antibiotic doxycycline for 4-6 weeks, proves the value of this endosymbiont as a therapeutic target¹⁰⁻¹⁶. Depletion of this endosymbiotic bacteria was confirmed in these trials and resulted in the permanent sterilisation of the adult worms, blocking parasite transmission, followed by a slow innocuous death of the adult worm over 12-24 months.'

The reviewer has also has an issue with the details of our time-kill assay 'The second minor criticism I have is of the 'fast acting' claim. Its not clear from the figure or the text whether equivalent concentrations were used, or what the temporal dynamics of the compound's effects are. Its also not clear whether the compounds can completely rid the cells of Wolbachia, and over what time periods clearing can occur. Maybe the dynamics are such that a 70% reduction is seen quicker with the new chemotypes, but that 100% elimination happens quicker with dox for example. Also, how much clearing of Wolbachia is needed to compromise the nematode life cycle? Addressing these issues one way or another would help improve the manuscript. '

For clarity the time-kill experiments were performed at 10 X EC₅₀ for each compound (*expanded detail included on page 8, see below*) and we determined the rate of depletion from Mf at day 6 after 1,2 or 6 days of drug exposure. In this assay the data is normalised to the vehicle control with maximum *Wolbachia* readout of 70-90% within the constraints of the assay. For clarity we have included the result for doxycycline exposure after 6 days in *Figure 5* to demonstrate that these novel compounds are equivalent to the maximal readout of this assay but within a 1-2 day exposure timeframe unlike the proof of concept drugs e.g. doxycycline.

In terms of clinical impact a series of proof of concept clinical trials have been performed with doxycycline, minocycline and rifampicin that empirically confirm that *Wolbachia* depletion greater than 90% in adult nematodes is sufficient to result in nematode death and all the other anti-*Wolbachia* features described in the introduction (Debrah *et al.*, 2007, 2011, 2015; Hoerauf *et al.*, 2008; Turner *et al.*, 2010). Therefore, there is strong experimental data that validates the translation of *in vitro* and *in vivo* *Wolbachia* depletion in a range of pre-clinical models with the desired clinical end point in terms of reduced *Wolbachia* load and nematode death.

Page 8:

'To compare the dynamics of the *in vitro* phenotype of our hits with existing anti-*Wolbachia* therapies, we employed a timed wash-out strategy (Figure 5) at 10x EC₅₀ to establish the time-kill kinetics of the selected chemotypes (1A to 5A). Time-kill kinetics of tetracyclines (doxycycline 1.21µM and minocycline 0.9µM) and rifampicin (0.2µM) indicated that it takes >2 days to achieve a maximal *Wolbachia* reduction in this assay (70-90% when normalised to vehicle control). In contrast, all hit chemotypes (1A to 5A) demonstrated this maximal activity (> 70% *Wolbachia* reduction) after just 2 days exposure (2.3µM ±0.7µM). Even more strikingly, after just a single day of exposure, doxycycline is inactive (9% *Wolbachia* reduction), yet chemotype 3A showed a 42% reduction and chemotypes 1A, 2A, 4A and 5A achieved *Wolbachia* reductions of 57 – 67%. This data presents a dramatic temporal shift in the exposure time of less than 2 days to achieve a maximal *Wolbachia* reduction for these novel chemotypes compared to greater than 4 days for known POC anti-*Wolbachia* drugs.

Reviewer #3

We are grateful to the reviewer for the very positive comments on our manuscript.

'This work uses cell based screen with heterologous expression of target coupled with counterscreens to identify new leads targeting Wolbachia. Overall the work is executed at the state of the art and the conclusions presented within the manuscript are well supported by the presented data. The screening campaign Identified around 1000 compounds with potency better than 1 uM that were nontoxic to mammalian cells. The group then used standard methods to focus down on fast acting compounds with good drug like properties, ultimately Identified several that are faster acting than doxycycline, more potent, and selective. This is a key and important finding – suggesting that there are multiple opportunities for optimization of drug candidates from this foundation.

The work is well written and clearly presented; there are no significant changes that need to be made.'

Reviewers' Comments:

Reviewer #2:

Remarks to the Author:

The authors have satisfactorily addressed my concerns.